# Holocene spatiotemporal millet agricultural patterns in northern China: A dataset of archaeobotanical macroremains

Keyang He[1], Houyuan Lu[1,2,3], Jianping Zhang[1,2], Can Wang[4]

[1]Key Laboratory of Cenozoic Geology and Environment, Institute of Geology and Geophysics, Chinese Academy of Sciences, Beijing, 100029, China
[2]Innovation Academy for Earth Science, Chinese Academy of Sciences, Beijing, 100029, China
[3]College of Earth and Planetary Sciences, University of Chinese Academy of Sciences, Beijing, 100049, China
[4]School of History and Culture, Shandong University, Jinan, 250100, China

*Correspondence to*: Keyang He (hekeyang1991@163.com)

**Abstract.** Millet agriculture, i.e., broomcorn millet (*Panicum miliaceum*) and foxtail millet (*Setaria italica*), were initially originated in northern China and provided the basis for the emergence of the first state in the Central Plains. However, owing to the lack of a comprehensive archaeobotanical dataset, when, where, and how these two millet types evolved across different regions and periods remains unclear. Here, we presented a dataset of archaeobotanical macroremains (n = 538) spanning the Neolithic and Bronze Ages in northern China and suggested a significant spatiotemporal divergence of millet agriculture in the subhumid mid-lower Yellow River (MLY) and semiarid agro-pastoral ecotone (APE). The key timing of the diffusion and transition of millet agriculture occurred around 6000 cal. a BP, coinciding with the Holocene Optimum (8000–6000 cal. a BP) and Miaodigou Age (6200–5500 cal. a BP). It spread westward and northward from the MLY to the APE and underwent a dramatic transition from low-yield broomcorn millet to high-yield foxtail millet. The combined influence of warm-wet climate, population pressure, and field management may have promoted the intensification, diffusion, and transition of millet agriculture around 6000 cal. a BP. Thereafter, the cropping patterns in the MLY were predominated by foxtail millet (~ 80%), while those in the APE emphasized on both foxtail (~ 60%) and broomcorn millet under a persistent drying trend since the mid-Holocene. This study provided the first quantitative spatiotemporal cropping patterns during the Neolithic and Bronze Age in northern China, which can be used for evaluating prehistoric human subsistence, discussing past human-environment interaction, and providing a valuable perspective of agricultural sustainability for the future. The dataset is publicly available at https://doi.org/10.5281/zenodo.6669730 (He et al., 2022)

## 1. Introduction

Broomcorn (*Panicum miliaceum*) and foxtail (*Setaria italica*) millets are among the world's oldest crops and were initially domesticated in northern China (Diao and Jia, 2017; Lu et al., 2009a; Yang et al., 2012; Zhao, 2011b), which played a significant role in the formation of the early Chinese civilization (Yuan et al., 2020; Zhao, 2011a) and prehistoric food globalization (d'Alpoim Guedes and Bocinsky, 2018; Dong et al., 2017; Jones et al., 2011; Liu et al., 2019), and remaining as

staple cereals in arid and semiarid regions. Although broomcorn and foxtail millet are generally similar in ecophysiology and are simultaneously excavated in archaeological sites in China (Dong et al., 2016; Zhao, 2011b), recent archaeobotanical studies suggested that the biogeography of the two millet types through time is distinctive (Hunt et al., 2008; Liu et al., 2009). In particular, when, where, and how these two millet types evolved across different regions and stages spanning the

Neolithic and Bronze Ages in northern China remains unclear.

Regarding the diachronic change in millets, the predominant crop during the Peiligang period (8000–7000 cal. a BP) was the broomcorn millet across the major centers of domestication, including Xinglonggou (Zhao, 2004), Dadiwan (Liu et al., 2004), Cishan (Lu et al., 2009a), Zhuzhai (Bestel et al., 2018; Wang et al., 2018a), and Yuezhuang sites (Crawford et al., 2016) (Figure 1), while the cropping patterns were dominated by the foxtail millet since the late Yangshao period (6000–

5000 cal. a BP) (Zhou et al., 2011), as evidenced by a series of archaeobotanical surveys in the Yiluo (Lee et al., 2007; Zhang et al., 2014), Ying (Fuller and Zhang, 2007; Zhang et al., 2010a), and Sushui valleys (Song et al., 2019). In contrast, phytolith evidence from the Guanzhong basin (Zhang et al., 2010b) and Central Plains (Luo et al., 2018; Wang et al., 2017; Wang et al., 2019; Zhang et al., 2012) indicated that the broomcorn millet-dominated cropping pattern persisted from the Peiligang to Erligang periods (3600–3300 cal. a BP). Whether and when there occurred a transition from broomcorn to

foxtail in northern China is still in dispute (Qin, 2012).

Compared with the intensive attention paid to the temporal changes in the broomcorn and foxtail millets, their spatial divergence has rarely been discussed. Crawford noted that broomcorn millet was traditionally more important in the drier western areas, while foxtail millet was more common in the east (Crawford et al., 2005). A review of cropping patterns in the mixed farming region suggested a selection of foxtail millet in the southward spread of millet agriculture to the mid-

upper Yangtze River (He et al., 2017). In addition, recent studies have demonstrated the north–south cropping patterns on the Loess Plateau, with a dominance of broomcorn millet in the north steppe area and foxtail millet in the south shrub-grassland area during the Longshan period (5000–3800 cal. a BP) (Bao et al., 2018; Sheng et al., 2018). However, most of these studies were confined to intraregional comparisons, and integrated research across different regions in northern China was unavailable.

To investigate the evolution of millet cropping patterns across different periods and regions, we compiled a comprehensive dataset of 538 flotation results from 381 sites spanning the Neolithic and Bronze Ages in northern China (Figure 1) and compared them with phytolith results obtained in our previous study. Based on the systematic analysis of crop data and demographic and climatic records, we revealed the spatiotemporal divergence between broomcorn and foxtail millet and the dynamic driving mechanism behind these phenomena.


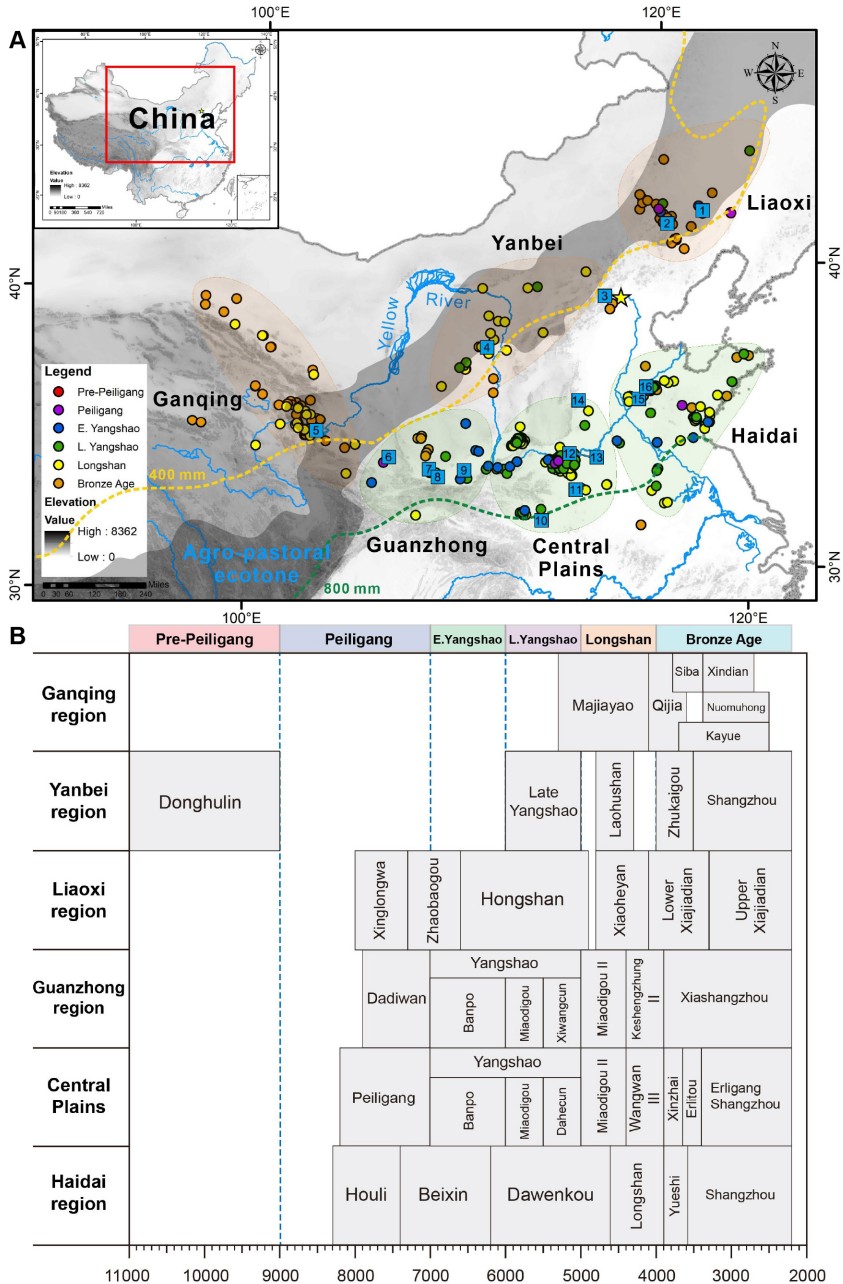

**Figure 1:** Distribution of archaeological sites of millet and related cultural sequence during the Neolithic and Bronze Ages in northern China. (A) The archaeological sites were classified into color-shaded subregions modified from Yan (2000). The gray shaded area and dashed lines indicate the agro-pastoral ecotone (APE) and annual average precipitation of 400 and 800 mm, respectively. The blue squares denote archaeological sites mentioned in this study: 1. Xinglonggou, 2. Weijiawopu, 3. Donghulin, 4. Shimao, 5. Lajia, 6. Dadiwan, 7. Wangjiazui, 8. Anban, 9. Yangguanzhai, 10. Baligang, 11. Jiahu, 12. Zhuzhai, 13. Dongzhao, 14. Cishan, 15. Yuezhuang, 16. Xihe. (B) Division of archaeological cultures and periods across the six subregions.

## 2. Archaeological background

Based on the spatial framework proposed by Yan (2000), the Neolithic cultures in northern China were subdivided into six subregions, i.e., Liaoxi, Yanbei, Ganqing, Guanzhong, Central Plains, and Haidai regions (Figure 1A). The first three regions are located in the semiarid agro-pastoral ecotone (APE) (Figure 1A) (Chen, 2018), while the last three regions are situated in the subhumid mid-lower Yellow River (MLY). The Liaoxi region is situated in the west of Liaoning Province and southeast of Inner Mongolia Autonomous Region; to the west, the Yanbei region is situated in the central-south of Inner Mongolia Autonomous Region and north of Shaanxi, Shanxi, and Hebei Provinces, northern parts of Chinese Loess Plateau; in the westmost, the Ganqing region is situated in the northeast of Qinghai Province and central-south of Gansu Province. The Guanzhong region is located in the southeast of Gansu Province and south of Shaanxi Province, which is merged into the Central Plains in some studies; to the east, the Central Plains is located in the south of Shanxi Province, and the bulk of Henan Province; the Haidai region is mainly located in the Shandong Province.

Regarding the temporal sequence in these regions, the Neolithic and Bronze cultures were summarized into six periods, including the Pre-Peiligang (11000–9000 cal. a BP), Peiligang (9000–7000 cal. a BP), early Yangshao (7000–6000 cal. a BP), late Yangshao (6000–5000 cal. a BP), and Longshan periods (5000–3800 cal. a BP) and the Bronze Age (3800–2221 cal. a BP) (Figure 1B) (The Institute of Archaeology and China Academy of Social Sciences, 2010). During the Pre-Peiligang period, only a few archaeological sites were scattered in the Yanbei region (i.e. Donghulin culture), Central Plains, and Haidai region. Subsequently, during the Peiligang period, four archaeological cultures, i.e. Xinglongwa, Dadiwan, Peiligang, and Houli, formed almost synchronously across Liaoxi, Guanzhong, Central Plains, and Haidai regions, which were widely regarded as early Neolithic cultures with sedentary settlements, origin of millet agriculture, pottery, and ground stone tool. During the early and late Yangshao period, settlements increased dramatically and spread wildly across all the six regions, especially Miaodigou culture, with remarkable signals of social hierarchy emergence. The Longshan period witnessed high population densities and the rise and fall of early complex society, and eventually the early states—Erlitou culture formed in the Central Plains in the Bronze Age (Liu and Chen, 2012).

## 3. Materials and methods

A total of 538 flotation results from 381 sites with millets excavated in northern China were assembled from a review of relevant literature, including published research papers, reports, and dissertations. Each site was assigned an age based on the direct dating *in situ* or the median age of archaeological culture. The following sites were eliminated: (1) sites belonging to historic periods beyond the study period and (2) sites with less than five grains of excavated staple crops, which may cause large randomness. After two screening steps, 487 flotation results from 349 sites were retained for further analysis.

Each flotation result addressed here indicated a compilation of original samples floated from the same cultural phase of an archaeological site. The original sample numbers investigated and compiled per cultural phase of each site ranged from 1 to 1082, with an average of approximate 30 samples. The 487 flotation results from 349 sites were classified into six

geographical regions and six cultural periods (Table 1), with 78 sites containing more than one single cultural phase. The number of flotation results across the six regions during each period was generally even, except for the common lack of flotation results in the agro-pastoral ecotone before Longshan period.

**Table 1.** Spatiotemporal composition of flotation results in northern China

| Period | Mid-lower Yellow River (MLY) | | | Agro-pastoral ecotone (APE) | | | |
|---|---|---|---|---|---|---|---|
| | Guanzhong region | Central Plains | Haidai region | Ganqing region | Yanbei region | Liaoxi region | Sum |
| Pre-Peiligang | 0 | 0 | 0 | 0 | 1 | 0 | 1 |
| Peiligang | 2 | 4 | 4 | 0 | 0 | 3 | 13 |
| Early Yangshao | 11 | 6 | 3 | 0 | 0 | 2 | 22 |
| Late Yangshao | 15 | 50 | 11 | 0 | 3 | 1 | 80 |
| Longshan | 11 | 72 | 29 | 22 | 26 | 0 | 160 |
| Bronze Age | 18 | 71 | 30 | 52 | 6 | 34 | 211 |
| Sum | 57 | 203 | 77 | 74 | 36 | 40 | 487 |

Five staple crops were summarized according to traditional Chinese agriculture (Liu et al., 2015): foxtail millet (*S. italica*), broomcorn millet (*P. miliaceum*), rice (*Oryza sativa*), wheat and barley (*Triticum aestivum* and *Hordeum vulgare*), and soybean (*Glycine max*). The flotation results were recounted with uniform standards referring to the concept of "number of

identified specialness taxon (NISP)" in zooarchaeology (Grayson, 2014), defined as the number of identified specimens for a specific site or skeleton. According to this criterion, fragments of unidentified crop seeds were not counted, while each fragment of identifiable large crop seeds, such as wheat and rice, that retained more than half of intact seed were counted as an intact seed; different parts of the crop seeds, i.e. diagnostic grains and spikelet bases of wheat, barley, and rice, retrieved from the same context were added up to denote the total numbers of crop seeds.

Considering the significant differences between the weight of the grains of crop species, the counts may not reflect their real status in the subsistence. Thus, they were converted into weights for better comparison of their actual values (Zhou et al., 2016b). The staple crops were converted into weights using the average weight of 1000 grains (Table 2) based on the Chinese Crop Germplasm Resources Information System to estimate the actual yield proportions.

Percentages (ratio/proportion of total crop remains) were applied to the counts and weights of the staple crops for each site

(Marston, 2014). Temporally, the sites were classified into six standardized periods and calculated for the mean values and standard deviations across the six regions (Figures 2 and 3). Besides, the percentage of the counts were summarized into the APE and MLY, which were plotted and smoothed using 5% "loess" regression method and 1000 bootstrap simulations in Acycle v2.4.1 (Figure 4) (Li et al., 2019). Spatially, the percentage of the counts of each site was plotted at intervals of 2000 or 1000 years and presented in graduated colors using ArcMap 10.6 (Figure 5).


**Table 2.** Statistics of weights of modern staple crops.

| Crops | Number of samples | Weight (g/1000 grains) | | | |
|---|---|---|---|---|---|
| | | Minimum | Maximum | Average | Standard deviation |
| *Setaria italica* | 3584 | 1.0 | 5.90 | 2.823 | 0.597 |
| *Panicum miliaceum* | 2071 | 0.6 | 10.0 | 6.38 | 1.44 |
| *Oryza sativa* | 26818 | 2.37 | 86.9 | 24.661 | 3.1889 |
| *Triticum aestivum* | 15810 | 8.1 | 59.9 | 33.72 | 8.21 |
| *Hordeum vulgare* | 6437 | 5.5 | 66.8 | 40.76 | 7.11 |
| *Glycine. max* | 3995 | 38 | 350 | 151.1 | 41.1 |

Detailed information is available at http://www.cgris.net/

## 4. Results

### 4.1 Evolution of cropping patterns in the MLY

During the Peiligang period, the millet cropping patterns in the Guanzhong region, Central Plains, and Haidai region were dominated by the broomcorn millet, as indicated by the percentage of the counts (100, 32, and 55%) and weights (100, 32, and 55%) (Figure 2). The relative low percentages of millets in the Central Plains (Figures 2B and E) and Haidai region (Figures 2C and F) was mainly attributed to the quantities of rice excavated in a few sites, such as the Baligang (Deng et al., 2015), Jiahu (Zhao and Zhang, 2009), and Xihe sites (Jin et al., 2014). With a decrease in rice from the Peiligang to the early

Yangshao period in the Central Plains and Haidai region, a steady rise occurred in the percentage of the counts (58 and 85%) and weights (62 and 88%) of broomcorn millet, which maintained an advantage over foxtail millet.

From the early Yangshao period to the Bronze age, the percentage of the counts and weights of broomcorn millet dramatically decreased from approximately 60 to 10%, while those of foxtail millet gradually increased from approximately 30 to 80% (Figure 2). Notably, during the late Yangshao period, foxtail millet surpassed broomcorn millet in most of the

percentages of the counts and weights in the Guanzhong region (69 vs. 29%, 51 vs. 41%, respectively); Central Plains (72 vs. 20%, 57 vs. 25%); and Haidai region (53 vs. 36%, 33 vs. 42%).

The dominant role of foxtail millet improved during the Longshan period, as indicated by the percentages of the counts and weights in the Guanzhong region (77 and 54%, respectively), Central Plains (75 and 50%), and Haidai region (55 and 24%). Accordingly, the percentage of the counts (10–17%) and weights (9–25%) of the broomcorn millet reduced in these regions.

Subsequently, the gap between the foxtail and broomcorn millet widened in the Bronze Age.

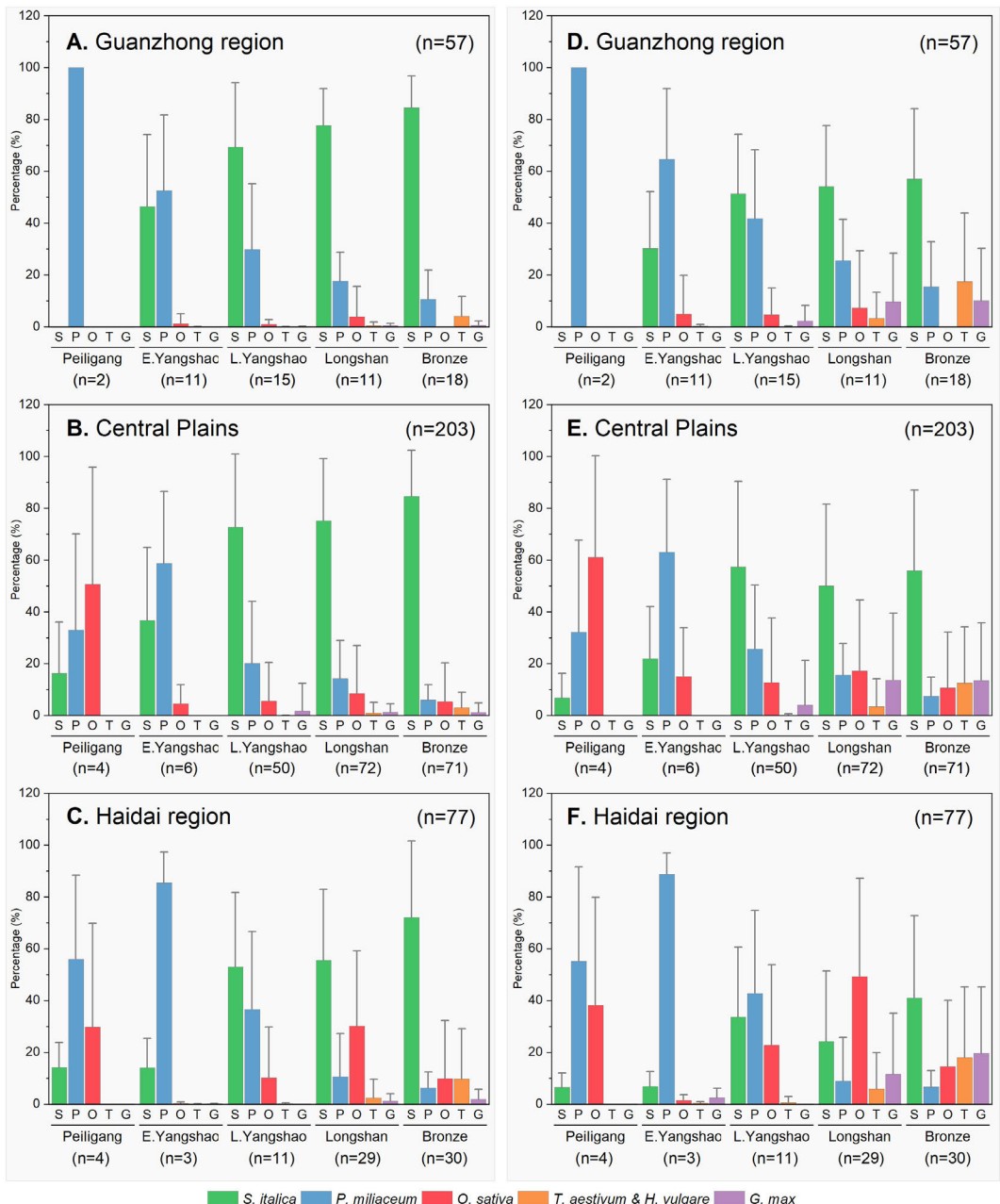

**Figure 2:** Evolution of cropping patterns in the subhumid mid-lower Yellow River (MLY). (A–C) The percentages of crop counts in the Guanzhong region, Central Plains, and Haidai region. (D–E) The percentages of crop weights in the Guanzhong region, Central Plains, and Haidai region.

**4.2 Evolution of cropping patterns in the APE**

The archaeobotanical studies in the APE were relatively rare and discontinuous, prior to the late Yangshao period. During the Pre-Peiligang period, the Donghulin site in the Yanbei region excavated the earliest charred millets in China, dominated by the foxtail millet (Zhao et al., 2020) (Figures 3B and E). Contrarily, the cropping pattern was dominated by the broomcorn millet during the Peiligang period in the Liaoxi region, accounting for 96 and 98% in the percentage of the counts and weights, respectively, particularly in the Xinglonggou site (Zhao, 2004) (Figures 3C and F). Subsequently, during the early Yangshao period, although the percentage of the counts and weights of the foxtail millet greatly increased to 40 and 26%, respectively, in the Liaoxi region (Figures 3C and F), particularly in the Weijiawopu site (Sun and Zhao, 2013), broomcorn millet still played a leading role in the cropping pattern.

From the late Yangshao period to the Bronze Age, the percentages of the counts and weights of broomcorn millet decreased in the Ganqing (45 to 30% and 55 to 27%, respectively); Yanbei (46 to 19% and 65 to 30%); and Liaoxi regions (96 to 35% and 98 to 48%) (Figure 3). The percentages of the counts and weights of foxtail millet gradually increased in the Yanbei (53 to 79% and 34 to 58%, respectively) and Liaoxi regions (3 to 64% and 1 to 50%), while the decrease in foxtail millet in the Ganqing region resulted from a rapid increase in wheat and barley (Figures 3A and D). Summarily, although the cropping patterns in the APE were dominated by foxtail millet during the Longshan period and Bronze Age, broomcorn millet was still significant.

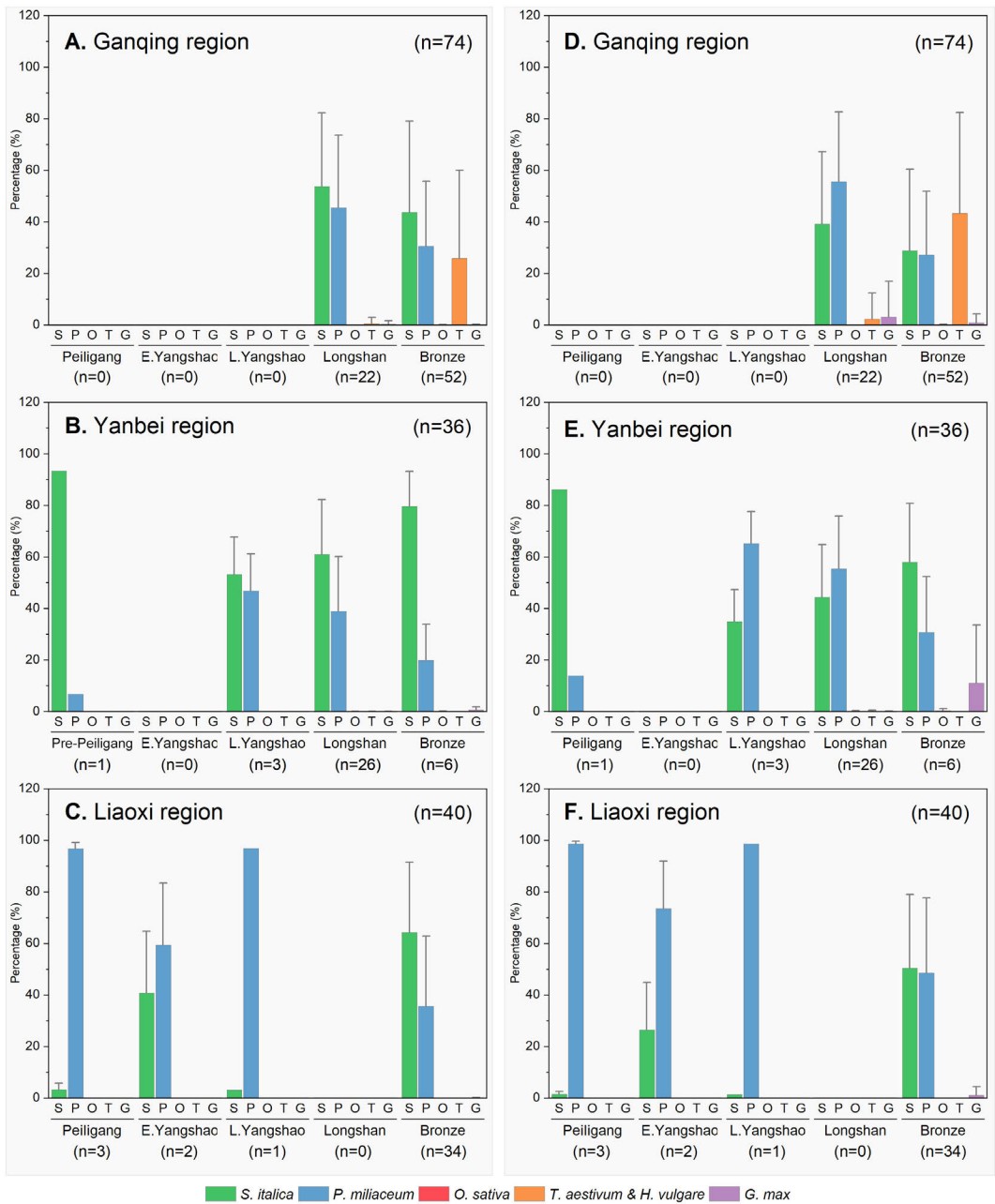

**Figure 3:** Evolution of cropping patterns in the semiarid agro-pastoral ecotone (APE). (A–C) The percentages of crop counts in the Ganqing, Yanbei, and Liaoxi regions. (D–E) The percentages of crop weights in the Ganqing, Yanbei, and Liaoxi regions.

### 4.3 Comparation of cropping patterns between the MLY and APE

The millet cropping patterns in northern China exhibited diverse patterns in the MLY and APE. In the MLY, the millet
cropping patterns strikingly transitioned from broomcorn to foxtail millet around 6000 cal. a BP (Figure 4A), such as the
Yangguanzhai site, the only settlement enclosed by a completed moat during the Miaodigou period (6000–5500 cal. a BP)
(Zhong et al., 2020). The percentage of the counts of foxtail millet rapidly increased from approximately 20 to 80% around
6000 cal. a BP and predominated afterward, and that of the counts of broomcorn millet changed in reverse (Figure 4A).
By contrast, although it appeared that the transition also occurred in the APE around 6000 cal. a BP despite the inadequate
data, the foxtail (~ 60%) and broomcorn millets (~ 40%) jointly made important contributions to the cropping patterns after
6000 cal. a BP (Figure 4B). For example, the staple crops in the Shimao site, a super-large central settlement dated 4300–
3800 cal. a BP, mainly comprised foxtail (73%) and broomcorn millets (26%) (Yang et al., 2022).

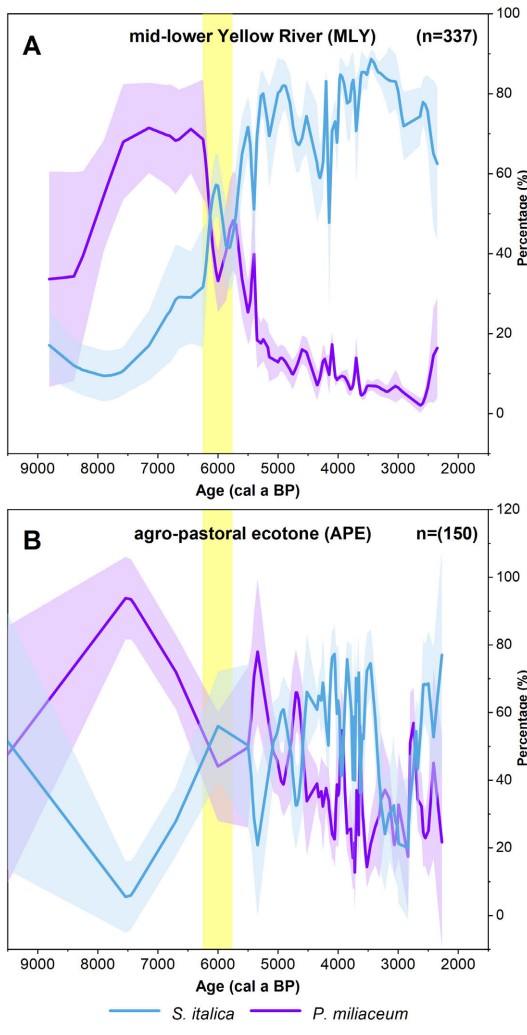

**Figure 4:** Transition of millet agriculture in the MLY (A) and APE (B). The percentages of foxtail and broomcorn millet counts are indicated in blue and purple, respectively. The shaded envelopes indicate the range of 1σ uncertainty of the percentage values. The yellow vertical bars denote the timing of transition around 6000 cal. a BP.

**4.4 Distribution of millet agriculture in northern China**

Based on the age, location, and percentage of crop counts of each site, the spatial distribution and temporal evolution of millet agriculture in northern China were divided into two stages.

Stage I (9000–6000 cal. a BP). Millet agriculture was confined to several possible centers of domestication along the marginal mountains of the loess plateau and Inner Mongolian plateau (Figures 5A–B and F–G), known as the Hilly flanks or China's Fertile Arc (Liu et al., 2009; Ren et al., 2016). The primary agricultural region in norther China had been framed at

stage I despite several blank areas still existed within this region. The predominant crop was broomcorn millet in most sites in the Liaoxi region, Haidai region, Central Plains, and Guanzhong region, generally accounting for over 60% (Figures 5F and G).

Stage II (6000–2221 cal. a BP). The discontinuous agricultural region linked together at stage II and expanded to marginal areas without solid evidence of agriculture prior to 6000 cal BP. The spread of millet intensified from the late Yangshao to

Longshan periods in two directions (Figures 5C–D and H–I): westward routine to the Ganqing region along the Wei River and Hexi corridor around 5400 cal BP (Leipe et al., 2019); and northward routine to the Yanbei region along the middle Yellow River around 5200 cal BP (Bao et al., 2018). The cropping patterns of most sites in northern China were now dominated by foxtail millet (mostly more than 60%), although a few sites in the APE were still dominated by broomcorn millet.


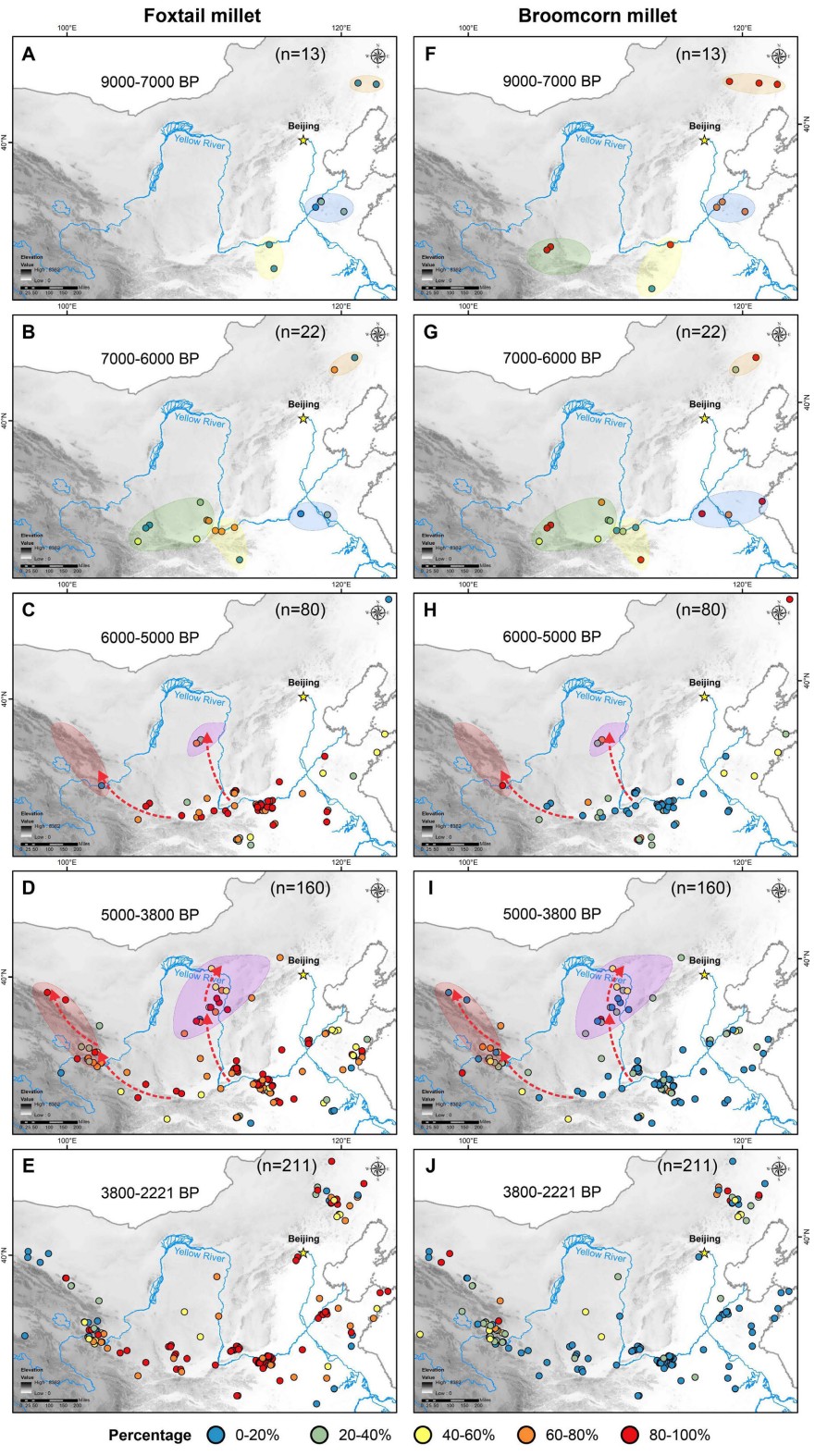

Foftail millet — Broomcorn millet

| A | (n=13) | 9000-7000 BP | F | (n=13) | 9000-7000 BP |
| B | (n=22) | 7000-6000 BP | G | (n=22) | 7000-6000 BP |
| C | (n=80) | 6000-5000 BP | H | (n=80) | 6000-5000 BP |
| D | (n=160) | 5000-3800 BP | I | (n=160) | 5000-3800 BP |
| E | (n=211) | 3800-2221 BP | J | (n=211) | 3800-2221 BP |

**Percentage**   ● 0-20%   ● 20-40%   ● 40-60%   ● 60-80%   ● 80-100%

**Figure 5:** Spatiotemporal distribution of millet sites along with cropping patterns in northern China. Graduated colors of the circles indicate the percentages of the counts of foxtail and broomcorn millets at each site. The light-colored shades denote the cultural subregions, and the red arrows indicate the spread routes of millet agriculture.


## 5. Discussion

### 5.1 Possible biases of archaeobotanical macroremains

Although foxtail and broomcorn millet could be identified based on clear diagnostic features of their carbonized seed (Liu and Kong, 2004) and phytolith morphologies (Lu et al., 2009b; Zhang et al., 2011), the percentage of the carbonized millet
appeared to exhibit a reverse proportion to those of the phytoliths in the archaeological context. The divergence between the macroremains and phytoliths may result from several factors, such as the different sources of crop grains and phytoliths or biases in the representativeness of quantity. Diagnostic phytoliths of millets were derived from the inflorescence bracts (Figures 6G and I), which may have been discarded during threshing and dehusking. Besides, based on the analysis of modern crop husk phytoliths, a previous study suggested that the counts of the husk phytoliths of foxtail and broomcorn
millet were proportional to the weights of their seed grains, which reflected the relative production rather than quantities of these two millet types (Zhang et al., 2010b). However, even after they were converted into weights, the percentage of carbonized millet still contradicted those of the phytoliths recovered from the same sites (Figures 6A–D), such as Anban (Liu, 2014; Zhang et al., 2010b), Lajia (Wang et al., 2015; Zhang, 2013; Zhao, 2003), Dongzhao (Luo et al., 2018; Yang et al., 2017), and Wangjiazui sites (Zhang et al., 2010b; Zhao and Xu, 2004). Thus, there may be other factors contributing to
this divergence, such as depositional and preservation biases.

Carbonized macroremains represented only a small and biased sample that had access to the fire, and the survival rates for the carbonization process varied among crops (Colledge and Conolly, 2014; Wright, 2003). Experiments on the effects of carbonization indicated that the carbonization temperature window of broomcorn millet (250–325 °C) was smaller than that of foxtail millet (270–390 °C), implying that broomcorn millet was less likely to be carbonized than foxtail millet in
archaeological contexts and thus may be underestimated in the cropping pattern (Märkle and Rösch, 2008; Wang and Lu, 2020). However, the preservation bias may be the reverse for phytoliths. Based on empirical evidence, the fragments of η-type phytoliths from broomcorn millet were generally larger than those of Ω-type phytoliths from foxtail millet (Lu et al., 2009b), implying that broomcorn millet was more likely to be preserved archaeologically and maintain diagnostic attributes for identification. In sum, the amount of broomcorn millet may be underestimated in carbonized macroremains and
overestimated in the phytolith assemblage. Thus, further taphonomic research on survival rates should be quantitatively conducted for the correction between foxtail and broomcorn millet.

Furthermore, the way of seed yield may also affect the biases between foxtail and broomcorn millet. Morphological data from the northern Chinese Loess Plateau showed that the size of broomcorn millet increased significantly during 5500–4000

cal BP, while that of foxtail millet didn't exhibit an obvious increasing trend (Bao et al., 2018). Given the contemporaneous

increasing human population, the crop yield was supposed to increase to feed a large population. Nevertheless, the increasing yields of broomcorn millet may depend on the increase in the seed size, while that of foxtail millet may result from the increase in the number of seeds per plant. Thus, the different ways of seed yield may also be one cause of the increase in the proportion of foxtail millet within the total seed number of seeds. In brief, both biases in the carbonization process and the way of seed yield may have exaggerated the proportion of foxtail millet.


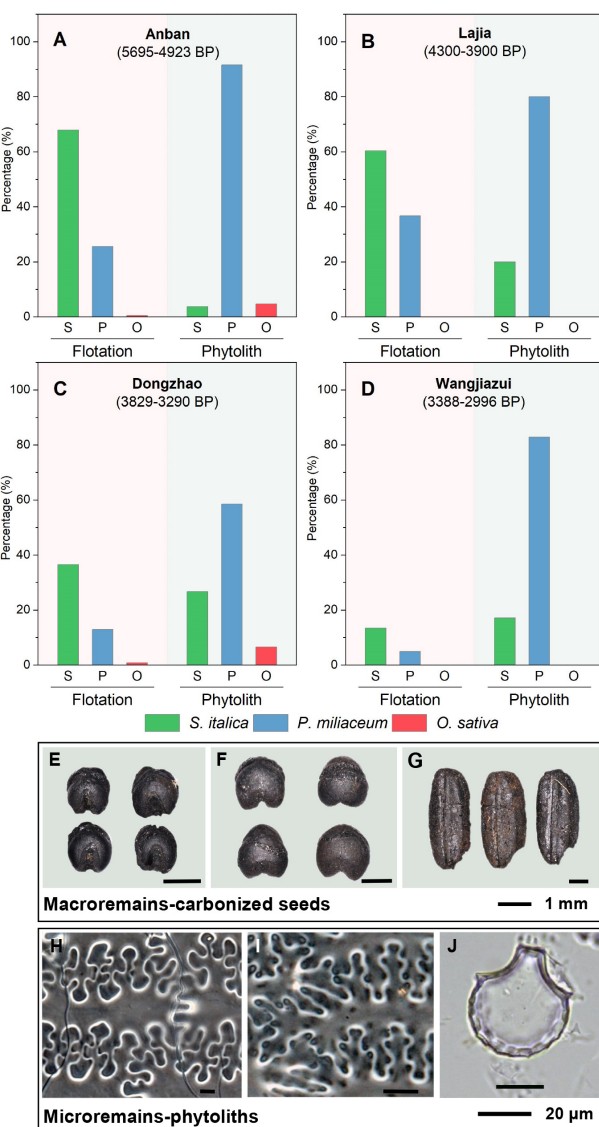

**Figure 6:** Divergence between flotation and phytolith results recovered from the same sites. (A) Anban (Liu, 2014; Zhang et al., 2010b), (B) Lajia (Wang et al., 2015; Zhang, 2013; Zhao, 2003), (C) Dongzhao (Luo et al., 2018; Yang et al., 2017), and (D) Wangjiazui (Zhang et al., 2010b; Zhao and Xu, 2004). The crop counts of the flotation results had been converted to weights before calculating the percentages.

(E–G) Images of charred seeds of foxtail (*S. italica*), broomcorn (*P. miliaceum*) millets, and rice (*O. sativa*), respectively. (H–J) Images of diagnostic phytoliths of foxtail (*S. italica*), broomcorn (*P. miliaceum*) millets, and rice (*O. sativa*), respectively.

## 5.2 Spatial divergence of cropping patterns in the MLY and APE

Though the transition from broomcorn to foxtail millet around 6000 cal. a BP occurred in both the MLY and APE (An et al., 2010; Li et al., 2021; Wang et al., 2016), the cropping patterns diversified afterwards to adapt to regional environments, which were predominated by foxtail millet in the MLY (Fuller and Zhang, 2007; Lee et al., 2007; Song et al., 2019; Yang et al., 2020; Zhang et al., 2014), and emphasized on foxtail and broomcorn millet in the APE (Bao et al., 2018; Sheng et al., 2018). The spatial divergence of the cropping patterns may be attributed to the crop traits and regional climate.

Compared to the foxtail millet, the broomcorn millet exhibited better resistance to saline-alkali soil and arid climates (mean annual precipitation: 400–700 mm vs. 300–500 mm) and required a shorter growing season and lower growing temperature constraints (mean annual temperature, 8–10 °C vs. 6–8 °C) (Shanxi Academy of Agricultural Sciences, 1987; Wang, 1996), and yet its yield was typically twice as low (1170 kg/hm2 vs. 750 kg/hm2) (Dong and Zheng, 2006). In sum, foxtail millet was more productive while the broomcorn millet was more environment tolerant.

Furthermore, though the East Asian monsoon rain belt had migrated northwestward ~ 150–300 km during the mid-Holocene (Dong et al., 2021; Yang et al., 2015), the persistent drying trend since the mid-Holocene in northern China (Goldsmith et al., 2017; Yang et al., 2021) may have formed diverse ecosystems and millet selection. Thus, under a gradual aridification trend since approximate 6000 cal. a BP, drought-tolerant broomcorn millet had been replaced by high-yield foxtail millet in the subhumid MLY but was still significant in the semiarid APE.

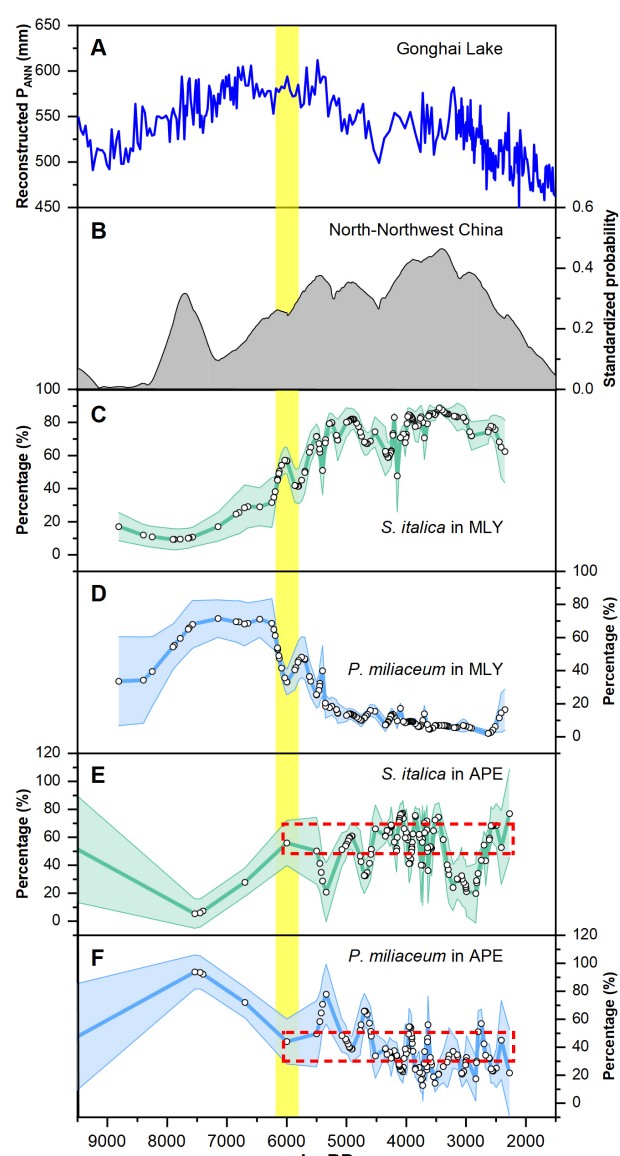

**Figure 7:** Comparison between cropping patterns and demographic and climatic records. (A) Pollen-based annual precipitation reconstructed from Gonghai Lake (Chen et al., 2015). (B) Demographic fluctuations inferred from the summed probability distribution of radiocarbon dates with 500-year smoothing in North and Northwest China (Wang et al., 2014). (C–F) Comparison of the percentages of foxtail (in green) and broomcorn (in blue) millet counts in the MLY and APE. The shaded envelopes indicate the range of 1σ uncertainty of the percentage values. The yellow vertical bars denote the high precipitation, population levels, and transition of millet in the MLY around 6000 cal. a BP. The red dashed rectangle indicates the roughly average values of millet in the APE after 6000 cal. a BP.

## 5.3 Transition and diffusion of millet agriculture in northern China

Based on the analysis of the archaeological and paleoclimatic records, it appeared that there occurred a diffusion, transition and divergence of millet agriculture in northern China around 6000 cal. a BP under the combined influence of climate change and cultural factors. The reasons are as follows.

First, although there was a dispute over whether a cooling or warming trend occurred in the late Holocene (Bova et al., 2021; Marsicek et al., 2018; Osman et al., 2021), global temperature anomalies reconstructed from a multi-proxy database of paleotemperature records exhibited ~ 0.6 °C of warming from 10000 to 6000 cal. a BP (Kaufman et al., 2020; Marcott et al., 2013), known as the Holocene thermal maximum (Renssen et al., 2009). In addition, pollen-based East Asian monsoon precipitation exhibited a time-transgressive pattern of the maximum precipitation shift from southern to northern China (Zhou et al., 2016a; Zhou et al., 2022), and peak precipitation occurred between 8000 and 5000 cal. a BP in northern China (Figure 7A), as indicated by pollen records in the Dali, Daihai, and Gonghai Lakes (Chen et al., 2015; Wen et al., 2017; Xiao et al., 2004). Thus, warm and wet climate during 8000–6000 cal. a BP with maximum combination of precipitation and temperature (Chen et al., 2015; Dong et al., 2022), defined as the Holocene optimum, may have greatly promoted the intensification of the millet agriculture shift from broomcorn millet to the productive foxtail millet in the MLY and the diffusion of millet agriculture from the subhumid MLY to the semiarid APE.

Second, the summed radiocarbon probability distribution suggested that the population in North and Northwest China experienced a rapid increase and reached a high level during 6500–5000 cal. a BP (Figure 7B) (Wang et al., 2014). Additionally, an explosive growth of archaeological sites occurred from the early (n = 217) to the late Yangshao periods (n = 3817) (Hosner et al., 2016), extensively distributed along the middle Yellow River (Figure 8). Consequently, the interaction among different cultural subregions unprecedentedly intensified since 6000 cal. a BP and formed the Chinese Interaction Sphere (Chang, 1986) or Miaodigou Age (6200–5500 cal. a BP) (Han, 2012). Thus, the cultural expansion and population explosion prompted the selection of the productive foxtail millet and the spread of millet agriculture around 6000 cal. a BP in northern China, coinciding with the origin and diffusion of the Sino-Tibetan evidenced by linguistics and genetics (Wang et al., 2021; Zhang et al., 2019).

Finally, as the main growing area of millet agriculture (He et al., 2017), Chinese loess was generally unsuitable for intensive arable production because of the small amounts of total nitrogen and phosphorus in the loess (Catt, 2001). Therefore, field management practices, particularly manuring, played significant roles in the intensification of millet agriculture in northern China. Modern field experiments suggested that the manuring (animal waste) of cereal crops could significantly elevate their $\delta^{15}N$ values above those grown in natural or unmanured soils (Bogaard et al., 2013), with an offset range of 5–7‰ for millet (Christensen et al., 2022). Elevated nitrogen isotope values of the millet grains recovered from the Baishui valley indicated the use of animal manure as a growth-promoting factor since the late Yangshao period (Wang et al., 2018b). Besides, a recent study in the Dadiwan site also suggested an intensive crop–livestock system was in practice around 5500 cal BP (Yang et al., 2022). Considering the yield of foxtail millet was typically twice that of broomcorn millet, the dramatic

increase in soil fertilization may promote the conscious choice of high-yield foxtail millets to maximize agricultural productivity.

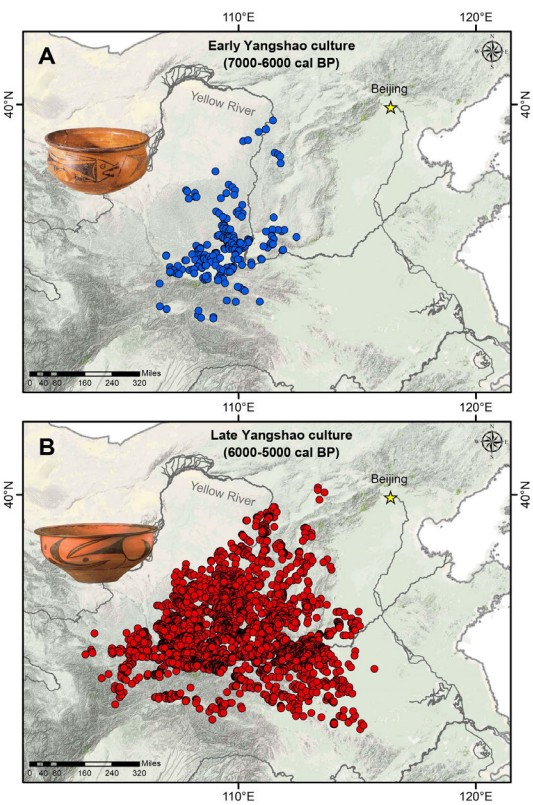

**Figure 8:** Distribution of the archaeological sites of Yangshao culture. (A) Distribution of archaeological sites during the early Yangshao period (7000–6000 cal. a BP). (B) Distribution of archaeological sites during the middle and late Yangshao periods (6000–5000 cal. a BP). Representative painted pottery basin with the fish design of the Banpo phase (A) and the petal design of the Miaodigou phase (B).

## 6. Data availability

Dataset of archaeobotanical macroremains including absolute counts and weights of staple crops, and percentage of counts and weights for each sites together with their locations and ages are available at the open-access repository Zenodo (He et al., 2022; https://doi.org/10.5281/zenodo.6669730)

## 7. Conclusion

Based on the synthesis of archaeobotanical macroremains spanning the Neolithic and Bronze Ages in northern China, our study determined the distinctive spatiotemporal patterns of millet agriculture in the MLY and APE. On the one hand, millet agriculture spread westward and northward from the MLY to the APE around 6000 cal. a BP. On the other hand, the cropping patterns diversified in the MLY and APE after the transition from broomcorn to foxtail millet around 6000 cal. a BP to adapt to regional environments. The diffusion and transition of millet agriculture in northern China around 6000 cal. a BP may have been driven by the combined influence of warm-wet climate, population pressure, and field management. Afterward, the persistent drying trend since the mid-Holocene may have formed the millet agriculture emphasized on the foxtail millet and drought-tolerant broomcorn millet in the APE.

### Author contributions

Conceptualization: HL, KH. Methodology: KH, CW, JZ. Investigation: KH, CW, JZ. Visualization: KH. Supervision: HL. Writing - original draft: KH. Writing - review & editing: KH, HL.

### Competing interests

The authors declare that they have no conflict of interest.

### Acknowledgments

We sincerely thank Xiaoshan Yu, Xue Yan, Xiaoqu Zheng, Yuqian Wang and Yongchao Ma for their assistance with data collection. We would like to thank Editage (www.editage.cn) for English language editing.

### Financial support

This research was supported by the National Natural Science Foundation of China (Nos. T2192954, 41830322, and 41902187), the Strategic Priority Research Program of the Chinese Academy of Sciences (Nos. XDB26000000), and China Postdoctoral Science Foundation (Nos. 2020M670444).

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
