# Peer review of "Holocene spatiotemporal millet agricultural patterns in northern China: A dataset of archaeobotanical macroremains"

_Earth System Science Data, 2022_

## Author Comment (AC1)

**Response to comments of Anonymous Referee #1**

**General comments**

The essay on the development of millet cultivation in China includes an interesting and large collection of data well suited for publication.

**Specific comments**

**1.** The temporal-quantitative evaluation is not comprehensible if the authors do not disclose how many sites (features, samples) they have per region and per time slice or archaeological culture. Only then is it clear whether the quantitative changes are not artifacts. According to page 4, they have 487 flotation results (are these samples?) from 349 sites. That is, less than 2 samples per site on average? Maybe also a few sites (which epochs) with many samples? Therefore, the representativeness of the data is not clear. What about the earliest time slice (e.g. Fig. 5 above): is there nothing investigated, or is it investigated, but nothing found?

**Response**: Thanks for the helpful comment. Sample numbers had been added to the top-right corner of figures per region (Fig. 2 and 3) and per time slice (Fig. 4). The 487 flotation results addressed in the manuscript indicate compilations of samples from the same phase of an archaeological site rather than original samples floated, and sites with less than 2 or many samples just indicate the number of cultural phases per site. Original sample numbers and volumes investigated per site have been added in the new version of the data tables (https://doi.org/10.5281/zenodo.6669730). The earliest time slice of Fig. 5 (11000–9000 cal BP) only consists of one site, i.e. Donghulin site (Fig. 3B), and thus is not illustrated again.

**2**. An image of selected macro-remains and phytoliths of the millets is missing. What are the criteria to distinguish the millets on the basis of their phytoliths? Is this possible? In any case, the comparison of grain numbers and phytoliths is not useful. If there has not been threshing and dehusking within the village, phytoliths will be hardly found on site.

**Response**: Images of selected macro-remains and phytoliths of the millets have been added in Fig. 6. Discrimination of millets phytoliths was based on five key diagnostic characteristics in phytolith morphology of inflorescence bracts, especially the Ω-type and η-type (Fig. 6H and I), which had been published by our research team (Lu et al., 2009). The divergence between grain numbers and phytoliths may attribute to several factors, and the possible effect of the processing method had been added to the discussion.

Lu, H. Y., et al. Phytoliths analysis for the discrimination of Foxtail millet (*Setaria italica*) and Common millet (*Panicum miliaceum*), PLoS One, 4, e4448, 2009a.

**Lines193–195**

…such as the different sources of crop grains and phytoliths or biases in the representativeness of

quantity. Diagnostic phytoliths of millets were derived from the inflorescence bracts (Figures 6G and I), which may have been discarded during threshing and dehusking. …

[Figure]

**Macroremains-carbonized seeds**                    2 mm

**Microremains-phytoliths**                    20 μm

**3**. Important would still be the climate discussion: around 6000 BP there is a climate deterioration in Central Europe, what it's like in China? The authors write warm/humid. How can they read this from the pollen data?

**Response**: According to the high-resolution, pollen-based quantitative precipitation reconstructed from the Gonghai Lake in northern China (Fig. 7A), the annual precipitation was 30% higher than present from ~7800–5300 ka, and no climate deterioration had been observed around 6000 cal BP in northern China. Besides, similar results could also be observed in pollen records from the Daihai and Hulun Lake.

**4**. Absolutely necessary is a chronology table, broken down by regions and millennia (better centuries), otherwise the arguments and data are not understandable. And it would be important to have a brief summary of what characterizes these archaeological cultures about which they are writing, see also Fig. 1. Are these comparable settlement types and types of findings?

**Response**: A chronology table broken down by regions and centuries has been added as Fig. 1B, and a brief summary of these archaeological cultures has been added in section 2.

**Line 76-82**

During the Pre-Peiligang period, only a few archaeological sites were scattered in the Yanbei region, Central Plains, and Haidai region. Subsequently, four archaeological cultures, i.e. Xinglongwa, Dadiwan, Peiligang, and Houli, formed almost synchronously across Liaoxi, Guanzhong, Central Plains, and Haidai regions, which were widely regarded as sedentary settlements and the origin of millet agriculture. During the early and late Yangshao period, settlements increased dramatically and spread wildly across all the six regions, especially Miaodigou culture, with remarkable signals of social hierarchy emergence. The Longshan period witnessed high population densities and the rise and fall of early complex society, and eventually the early states—Erlitou culture formed in the Central Plains during the Bronze Age.

[Figure]

B

| | Pre-Peiligang | Peiligang | E.Yangshao | L.Yangshao | Longshan | Bronze Age |

**5**. That more existing 14C data of settlements means an increase in population is an old idea but not convincing. The amount of 14C data depends among others on how much money the archaeologists spend on it. It is enough argument that there are more sites. But for that you would have to know whether the fewer, older sites are just as easy to find. If they have left fewer traces (e.g. block construction of houses, no pits), then you will also find less. Are all epochs sampled and examined equally (see above?).

**Response**: Thanks for the comment. One key process in the use of 14C data of settlements dealing with the inter-site sampling intensity was aggregating samples from the same site before calculating summed probability distribution, and thus may diminish possible anthropogenic biases and reflect the fluctuation of population. The taphonomic loss was an inevitable question in the flotation results, and feature numbers (mean, standard deviation) (Excel data, MLY_Fig.2 and APE_Fig.3 sheets) had been applied to infer changes in all epochs.

**6**. There is also a lack of inclusion of other crops. For example, a change from dry to wet rice cultivation from the Neolithic to the Bronze Age can be expected, etc. This would clearly substantiate the author's thesis of increasing effectiveness. In addition, the archaeological background would be interesting, what do we know: settlement concentration? Which raw materials? Already metal? Trade?

**Response**: Thanks for your suggestion. Due to limited research on the determination of rice arable systems, whether the paddy fields in northern China were wet (e.g. Huizui site) or dry (e.g. Zhuzhai site) remains controversial and a shift from dry to wet rice cultivation had not been observed so far. The increasing effectiveness was supposed to be mainly focused on millet agriculture in northern China as rice only played a minor role in the cropping patterns. The key change in cropping patterns

from the Neolithic to the Bronze Age was the introduction of wheat and barley through the trans-Eurasian exchange, as illustrated in Figs. 2 and 3.

**7.** There is something strange with nitrogen and loess on p. 15 below. The black soils from loess have been the most fertile soils ever, which probably did not have to be fertilized for the first 1-2 millennia.

**Response**: Though the black soils from loess were fertile, loess soils were vulnerable to the loss of organic matter due to the little content of clay in loess and thus may not be able to sustain continuous intensive cultivation. Besides, a millet–pig system illustrating the fertilization of millet fields with pig or human dung was supposed to be in practice at Dadiwan around 5500 cal BP in a recent study (Yang et al., 2022).

Yang, J. S., et al. Sustainable intensification of millet–pig agriculture in Neolithic North China, Nature Sustainability, https://doi.org/10.1038/s41893-022-00905-9, 2022.

**8**. In the case of bar charts, the dashed lines must be removed or consistently applied to everything. Mathematically, they are strange, because these are different times and data sets. Why are some bars missing in the charts? The number of the figures has to be checked.

**Response**: The horizontal dashed lines in Fig. 3 have been removed and the green and blue dashed lines have been plotted consistently to illustrate the diachronic trends of increase or decrease across different periods. The missing bars in the charts indicate the absence of certain crops or lack of flotation results during certain periods (Fig. 3). The number of Fig. 4 was mistakenly written as Fig. 2 and has been revised.

**9**. Something important for the calculations is the differentiation of mass finds (charred storage finds) versus normal settlement waste. If such mass finds, which are singular events, are included in the calculations, they confuse the results, e.g. by pretending an "increase".

**Response**: Thanks for the comment. Two rules applied in this study could ensure the representativeness of these data. Firstly, most of the flotation results compiled here were a mixture of contemporaneous samples recovered from cultural layers or pits of the same sites, with more than 74% containing 2 or more samples. Secondly, the absolute numbers of crops from each site were transferred to percentage first and then calculated for the mean values and standard deviations of each period, which diminished possible representativeness bias induced by samples with mass finds.

**Technical corrections**

**10**. As for the data tables (Excel):

They are not understandable for people not involved in the project.

The feature numbers, sample numbers and sample volumes investigated per site are lacking.

Are all items preserved charred?

The order of the data is unclear.

The archaeological period and the publication per site are lacking.

**Response**: As for the data tables (Excel): A new version of V1.0 has been uploaded, including a revised Excel file and a new KML file for better visualization of these data. Sample numbers and volumes of each flotation results have been added and feature numbers of each period (MLY_Fig.2 and APE_Fig.3 sheets) have also been illustrated. All staple crops are preserved charred. The order of the data has been rearranged. The archaeological period and the publication per site have also been added (https://doi.org/10.5281/zenodo.6669730).

---

## Author Comment (AC2)

**Response to comments of Anonymous Referee #2**

**General comments**

Millet agriculture were initially domesticated in northern China and played an important role in early agriculture evolution and the formation of the Chinese civilization. The manuscript reports a dataset of archaeobotanical macroremains spanning the Neolithic and Bronze Ages in northern China. Authors also suggest a significant spatiotemporal divergence of millet agriculture, discuss the past human-environment interaction, and provide a valuable perspective of agricultural sustainability for the future. This manuscript meets the scope of Earth System Science Data and could arise a wide audience as well. I would like to suggest a publication after a moderate revision.

**Response**: Thanks for the helpful comment. Our point-by-point responses are provided below.

**Specific comments**

**1.** In Introduction Part and Fig. 1. I suggest that authors check the names of different regions in North China, which belong to geographical division or archaeological culture division. For example, many archaeological sites distribute in Loess Plateau and are not in Guanzhong basin. Yanbei region is not clear.

**Response**: The reference for the names of different regions has been revised, and a brief introduction to the geographical and archaeological division of the six subregions has been added in Lines 73–79. Archaeological sites distributed on the Chinese Loess Plateau were divided into the Yanbei region in the north and Guanzhong region in the south by bondary of 37°N.

**Lines 73–79**

The Liaoxi region is situated in the west of Liaoning Province and southeast of Inner Mongolia Autonomous Region; to the west, the Yanbei region is situated in the central-south of Inner Mongolia Autonomous Region and north of Shaanxi, Shanxi, and Hebei Provinces, northern parts of Chinese Loess Plateau; in the westmost, the Ganqing region is situated in the northeast of Qinghai Province and central-south of Gansu Province. The Guanzhong region is located in the southeast of Gansu Province and south of Shaanxi Province, which is merged into the Central Plains in some studies; to the east, the Central Plains is located in the south of Shanxi Province, and the bulk of Henan Province; the Haidai region is mainly located in the Shandong Province.

**References**

Yan, W. M.: Cradle of Oriental Civilization. In: The origins of agriculture and rise of civilization, Yan, W. M. (Ed.), Science Press, Beijing, 2000.

**2**. A reference (Zhou et al., 2011) need to be cited which has discussed the significant divergence of millet west Loess Plateau around 5500 BP.

**Response**: A new reference (Zhou et al., 2011) has been added in Line 40.

**References**

Zhou, X. Y., Li, X. Q., Zhao, K. L., Dodson, J., Sun, N., and Yang, Q.: Early agricultural development and environmental effects in the Neolithic Longdong basin (eastern Gansu), Chinese Sci. Bull., 56, 762, https://doi.org/10.1007/s11434-010-4286-x, 2011.

**3**. In Discuss Part.

Line 170-175

The description on "The spread of millet intensified from the late Yangshao to Longshan periods in two directions (Figures 5C–D and H–I): westward routine to the Ganqing region and northward routine to the Yanbei region" and Fig. 5 need more evidences and the references to support

**Response**: More evidence of routes and dates in the two directions had been added in Lines 188–192, and new references supporting the spread of millet agriculture had also been added.

**Lines 188–192**

The discontinuous agricultural region linked together at stage II and expanded to marginal areas without solid evidence of agriculture prior to 6000 cal BP. The spread of millet intensified from the late Yangshao to Longshan periods in two directions (Figures 5C–D and H–I): westward routine to the Ganqing region along the Wei River and Hexi corridor around 5400 cal BP (Leipe et al., 2019); and northward routine to the Yanbei region along the middle Yellow River around 5200 cal BP (Bao et al., 2018).

**References**

Bao, Y. G., Zhou, X. Y., Liu, H. B., Hu, S. M., Zhao, K. L., Atahan, P., Dodson, J., and Li, X. Q.: Evolution of prehistoric dryland agriculture in the arid and semi-arid transition zone in northern China, PLoS One, 13, e0198750, https://doi.org/10.1371/journal.pone.0198750, 2018.

Leipe, C., Long, T. W., Sergusheva, E. A., Wagner, M., and Tarasov, P. E.: Discontinuous spread of millet agriculture in eastern Asia and prehistoric population dynamics, Sci. Adv., 5, eaax6225, https://doi.org/10.1126/sciadv.aax6225, 2019.

**4**. Line 195-210

The discussion on the possible biases of archaeobotanical macroremains and the reason of divergence of the foxtail and broomcorn millet need to add some information on the different ecological habits and the way of seed yield from the foxtail and broomcorn millet. I think that the discussion of phytolith and Fig. 6 are not necessary, which can't support the changes and divergence of the foxtail and broomcorn millet during the Neolithic.

**Response**: Different ecological habits and the seed yield of foxtail and broomcorn millets have been added in Lines 227–234. Though discussion of phytolith (Fig. 6) didn't support the transition from broomcorn to foxtail millet, it could also provide a valuable perspective for comparison to that of charred seeds.

**Lines 227–234**

Furthermore, the way of seed yield may also affect the biases between foxtail and broomcorn millet. Morphological data from the northern Chinese Loess Plateau showed that the size of broomcorn millet increased significantly during 5500–4000 cal BP, while that of foxtail millet didn't exhibit an obvious increasing trend (Bao et al., 2018). Given the contemporaneous increasing human population, the crop yield was supposed to increase to feed a large population. Nevertheless, the increasing yields of broomcorn millet may depend on the increase in the seed size, while that of foxtail millet may result from the increase in the number of seeds per plant. Thus, the different ways of seed yield may also be one cause of the increase in the proportion of foxtail millet within the total seed number of seeds. In brief, both biases in the carbonization process and the way of seed yield may have exaggerated the proportion of foxtail millet.

**5**. Line 259-269

The manuring enhanced the crop yields and provide the possible reasons that human adapt the environmental changes and can't well understand the divergence of the foxtail and broomcorn millet around 6000 BP. Authors need to more discussions on the driving factors.

**Response**: More discussions about the relationship between field manuring practices and different crop yields have been added in Lines 297–301. Given the yield of foxtail millet was typically twice that of broomcorn millet, the ancient human may choose to cultivate more foxtail millet to maximize crop yield on the condition of enough fertilizer.

**Lines 297–301**

Besides, a recent study in the Dadiwan site also suggested an intensive crop–livestock system was in practice around 5500 cal BP (Yang et al., 2022). Considering the yield of foxtail millet was typically twice that of broomcorn millet, the dramatic increase in soil fertilization may promote the conscious choice of high-yield foxtail millets to maximize agricultural productivity.

**References**

Yang, J. S., Zhang, D. J., Yang, X. Y., Wang, W. W., Perry, L., Fuller, D. Q., Li, H. M., Wang, J., Ren, L. L., Xia, H., Shen, X. K., Wang, H., Yang, Y. S., Yao, J. T., Gao, Y., and Chen, F. H.: Sustainable intensification of millet–pig agriculture in Neolithic North China, Nat. Sustain. https://doi.org/10.1038/s41893-022-00905-9, 2022.

---

## Referee Report (RR1)

The manuscript has been worked on by the authors but there are still most questions left unanswered mentioned already in my first review.

Abstract: "around 6000 cal. a BP, coinciding with the Holocene optimum….". The Holocene optimum in the 6th millennium was cal BC not BP ((the latter is around 4000 BC)). In addition, the cultivated plants (*Triticum* species, *Hordeum*, *Secale*, pulses, etc.) in South-West Asia (Near East) are millennia older.

Fig. 1: Different colored dots are entered in the map, which seem to correspond to regions. Are the dots of one color the same age? Otherwise, it would make more sense to present the points according to the archaeological epochs. Looking at the chronology table below, archaeological cultures are already present well before 6000 BP (see also in the text rows 80-90). Do we know nothing about these cultures archaeobotanically? Is this really a Neolithic (by what proved?) or are these hunter-gatherer cultures? What characterizes and differentiates the different archaeological cultures that are compared in Figs 2 and 3?

Line 100: the archaeozoological NISP method used for counting crop fragments is not understandable here. This should be clarified.

Line 102ff.: Perhaps the different size of the grains of *Triticum*, *Panicum*, *Setaria* etc. is meant here? This should be clarified.

Fig. 2 and 3: the dashed lines are mathematically not correct (see my last review), the percentages of counts and weights show the same trend.
The archaeobotanical results cannot be understood without the following (citation from my first review already):
"The temporal-quantitative evaluation is not comprehensible if the authors do not disclose how many sites (features, samples) they have per region and per time slice or archaeological culture. Only then is it clear whether the quantitative changes are not artifacts. According to page 4, they have 487 flotation results (are these samples?) from 349 sites. That is, less than 2 samples per site on average? Maybe also a few sites (which epochs) with many samples? Therefore, the representativeness of the data is not clear. What about the earliest time slice (e.g. Fig. 5 above): is there nothing investigated, or is it investigated, but nothing found?"

The phytololith picture of millets (Fig. 6 below to the left) is not good enough to be distinguished from the one to the middle and they have to be named by the species name. The grains have to be turned (embryo has to be below) and the pictures must be larger, to see if the curves of the embryos are typical for *Panicum* and *Setaria* respectively which are different.

I apologize but it makes no sense to comment on this draft further without the lacking fundamental information. I would like to ask the authors to look at my first review and to enter their answers to the open questions into their manuscript, please. I just will be able to have a look at a further fully corrected version of the manuscript with the changes indicated in the text.

---

## Author Response (AR2)

**Response to comments of Anonymous Referee #1**

**General comments**
The manuscript has been worked on by the authors but there are still most questions left unanswered mentioned already in my first review.

**Response:** Thanks for all the constructive comment. We have revised the manuscript carefully and our point-by-point responses are provided below.

**Q1**

Abstract: "around 6000 cal. a BP, coinciding with the Holocene optimum….". The Holocene optimum in the 6th millennium was cal BC not BP ((the latter is around 4000 BC)). In addition, the cultivated plants (*Triticum* species, *Hordeum*, *Secale*, pulses, etc.) in South-West Asia (Near East) are millennia older.

**Response:** Thanks for the helpful comment.
**1)** We have revised the age of Holocene optimum as intervals between 8000 and 6000 cal. a BP. Further clarification of the Holocene optimum, defined as maximum combination of precipitation and temperature (Chen et al., 2015; Dong et al., 2022), was added in lines 302–304.
**2)** Yes, we acknowledged that the domestication of crops in Southwest Asia occurred in the early Holocene (~9000 cal. a BP), which was millennia older than the millets in East Asia.

**References:**
Chen, F. H., Xu, Q. H., Chen, J. H., et al.: East Asian summer monsoon precipitation variability since the last deglaciation, Sci. Rep., 5, 11186, 2015.
Dong, Y. J., Wu, N. Q., Li, F. J., et al.: The Holocene temperature conundrum answered by mollusk records from East Asia, Nat. Commun., 13, 5153, 2022.

**Q2**

Fig. 1: Different colored dots are entered in the map, which seem to correspond to regions. Are the dots of one color the same age? Otherwise, it would make more sense to present the points according to the archaeological epochs. Looking at the chronology table below, archaeological cultures are already present well before 6000 BP (see also in the text rows 80-90). Do we know nothing about these cultures archaeobotanically? Is this really a Neolithic (by what proved?) or are these hunter-gatherer cultures? What characterizes and differentiates the different archaeological cultures that are compared in Figs 2 and 3?

**Response:** Thanks for the suggestion.
**1)** These dots of one color in the original Fig. 1A were not the same age and all the dots have been redrawn according to the archaeological epochs, which corresponded to the chronology table in Fig. 1B.
**2)** The archaeological cultures before 6000 BP have also been archaeobotanically investigated and exhibited as "Pre-Peiligang, Peiligang and Early Yangshao" periods in Figs. 2 and 3.
**3)** These cultures were proved to be Neolithic mainly based on the sedentary settlements, possible agriculture, pottery, and ground stone tool (Liu and Chen, 2012), and the characterizes and differentiates of these cultures were briefly summarized in lines 85–92.

**References:**

Liu, L. and Chen, X. C.: The archaeology of China: from the late Paleolithic to the early Bronze Age, Cambridge University Press, New York, 2012.

[Figure]

**Q3**

Line 100: the archaeozoological NISP method used for counting crop fragments is not understandable here. This should be clarified.

Line 102ff.: Perhaps the different size of the grains of *Triticum*, *Panicum*, *Setaria* etc. is meant here? This should be clarified.

**Response:** The definition of archeozoological NISP and possible effects of different size of crop grains in identification have been clarified in lines 111–115.

**Line 111–115**

*…(NISP)" in zooarchaeology (Grayson, 2014), defined as the number of identified specimens for a specific site or skeleton. According to this criterion, fragments of unidentified crop seeds were not counted, while each fragment of identifiable large crop seeds, such as wheat and rice, that retained more than half of intact seed were counted as an intact seed; different parts of the crop seeds, i.e. diagnostic grains and spikelet bases of wheat, barley, and rice, retrieved from the same context were added up to denote the total numbers of crop seeds.*

**Q4**

Fig. 2 and 3: the dashed lines are mathematically not correct (see my last review), the percentages of counts and weights show the same trend.

The archaeobotanical results cannot be understood without the following (citation from my first review already):

"The temporal-quantitative evaluation is not comprehensible if the authors do not disclose how many sites (features, samples) they have per region and per time slice or archaeological culture. Only then is it clear whether the quantitative changes are not artifacts. According to

page 4, they have 487 flotation results (are these samples?) from 349 sites. That is, less than 2 samples per site on average? Maybe also a few sites (which epochs) with many samples? Therefore, the representativeness of the data is not clear. What about the earliest time slice (e.g. Fig. 5 above): is there nothing investigated, or is it investigated, but nothing found?"

**Response:** Thanks for the insightful comment.
**1)** The dashed lines in Figs. 2 and 3 have been completely deleted in the manuscript.
**2)** Numbers of flotation results (sites or cultural phases) applied to illustrate the cropping patterns per region and per time slice have been added below each period in revised Figs. 2 and 3. Further detailed information on the spatiotemporal composition of flotation results have been added in Table 1 and lines 99–104.
**3)** 78 of 349 sites contained more than one cultural phase, while the rest contained one single cultural phase.
**4)** Only one site with millets were investigated in the earliest time slice of Fig. 5 (11000–9000 cal BP), i.e., Donghulin site (Fig. 3B), and thus were not illustrated again.

**Line 99–104**

*Each flotation result addressed here indicated a compilation of original samples floated from the same cultural phase of an archaeological site. The original sample numbers investigated and compiled per cultural phase of each site ranged from 1 to 1082, with an average of approximate 30 samples. The 487 flotation results from 349 sites were classified into six geographical regions and six cultural periods (Table 1), with 78 sites containing more than one single cultural phase. The number of flotation results across the six regions during each period was generally even, except for the common lack of flotation results in the agro-pastoral ecotone before Longshan period.*

**Table 1.** *Spatiotemporal composition of flotation results in northern China*

| Period | Mid-lower Yellow River (MLY) | | | Agro-pastoral ecotone (APE) | | | |
|---|---|---|---|---|---|---|---|
| | Guanzhong region | Central Plains | Haidai region | Ganqing region | Yanbei region | Liaoxi region | Sum |
| Pre-Peiligang | 0 | 0 | 0 | 0 | 1 | 0 | 1 |
| Peiligang | 2 | 4 | 4 | 0 | 0 | 3 | 13 |
| Early Yangshao | 11 | 6 | 3 | 0 | 0 | 2 | 22 |
| Late Yangshao | 15 | 50 | 11 | 0 | 3 | 1 | 80 |
| Longshan | 11 | 72 | 29 | 22 | 26 | 0 | 160 |
| Bronze Age | 18 | 71 | 30 | 52 | 6 | 34 | 211 |
| Sum | 57 | 203 | 77 | 74 | 36 | 40 | 487 |

**Q5**

The phytolith picture of millets (Fig. 6 below to the left) is not good enough to be distinguished from the one to the middle and they have to be named by the species name. The grains have to be turned (embryo has to be below) and the pictures must be larger, to see if the curves of the embryos are typical for *Panicum* and *Setaria* respectively which are different.

**Response:** Both the images of charred millet seeds and phytoliths have been rearranged, which were large and clear to be identified with diagnostic characteristics, i.e., the grain embryos of millets have been turned to be below (E and F) and the Ω- and η- undulated epidermal long cell walls (H and I) have been illustrated. Besides, they have all named by the Latin species name.

---

## Author Response (AR3)

**Response to comments of Anonymous Referee #1**

**General comments**

I suggest to replace the abbreviations in the summary by the real terms. Readers might read this part first and will not understand the content.

**Response:** Thanks for the helpful suggestion. The abbreviations in the summary have been changed to the real terms in lines 17–18 and 21–22.